# Severe Fever with Thrombocytopenia Syndrome with Necrotizing Lymphadenitis in a Patient who Underwent ^18^F-FDG PET/CT: A Case Report

**DOI:** 10.3390/ijerph181910488

**Published:** 2021-10-06

**Authors:** A Reum Kim, Taehwa Kim, Dong-Hoon Shin, Sujin Lee, Seungjin Lim

**Affiliations:** 1Division of Infectious Diseases, Department of Internal Medicine, Pusan National University Yangsan Hospital, Yangsan 50612, Korea; ar6493@naver.com (A.R.K.); beauty192@hanmail.net (S.L.); 2Division of Pulmonology, Allergy and Critical Care Medicine, Department of Internal Medicine, Pusan National University Yangsan Hospital, Yangsan 50612, Korea; taehwagongju@naver.com; 3Department of Pathology, Pusan National University Yangsan Hospital, Yangsan 50612, Korea; donghshin@chol.com; 4Research Institute for Convergence of Biomedical Science and Technology, Pusan National University Yangsan Hospital, Yangsan 50612, Korea

**Keywords:** severe fever with thrombocytopenia syndrome, lymphadenitis, positron-emission tomography, computed tomography

## Abstract

Severe fever with thrombocytopenia syndrome (SFTS), also known as fever, thrombocytopenia, and leukopenia syndrome, is an emerging tick-borne infectious disease in Asian countries. SFTS should be suspected in patients presenting with fever, thrombocytopenia, and leukopenia and have a history of tick exposure in an endemic area. Since SFTS can be accompanied by lymphadenopathy, particularly generalized lymphadenopathy, it can be confused with a primary malignancy, such as lymphoma. The study reports a case of SFTS accompanied by multiple lymphadenopathies, which mimicked malignant lymphoma on F-18 fluorodeoxyglucose positron emission tomography/computed tomography.

## 1. Introduction

Severe fever with thrombocytopenia-causing phlebovirus has been officially named *Dabie bandavirus* and belongs to the genus bandavirus, family *Phenuiviridae,* and order *bunyavirales* [1,2]. This virus is also known as the severe fever with thrombocytopenia syndrome virus (SFTSV). Severe fever with thrombocytopenia syndrome (SFTS) is an endemic tick-borne disease in China, South Korea, Japan, and Vietnam [1,2,3]. SFTS is caused by SFTSV and is most often transmitted by tick-bites (*Haemaphysalis longicornis* or *Amblyomma testudinarium*) [1]; person-to-person transmission can occur through blood or body fluids of an infected person [2]. Since SFTS can be accompanied by lymphadenopathy, particularly generalized lymphadenopathy, it can be confused with a primary malignancy, such as lymphoma. We report a case of SFTS accompanied by multiple lymphadenopathies, which mimicked malignant lymphoma on F-18 fluorodeoxyglucose positron emission tomography/computed tomography (^18^F-FDG PET/CT).

## 2. Case Report

A 60-year-old woman without known comorbidities was transferred to our hospital due to fever and abnormal laboratory test findings. She had not taken any medications prior to the emergence of symptoms. She had climbed a mountain about two weeks prior to our hospital admission and had no memory of insect or tick bite. The patient was admitted to another hospital due to fever after 7 days of climbing. She had a fever with a maximum temperature of 39 °C, aggravating nausea, and vomiting. Abdominal CT revealed hepatosplenomegaly. Therefore, she was referred to our hospital seven days after the onset of symptoms. The patient was alert with a blood pressure of 90/60 mmHg, pulse of 84 beats/min, and body temperature of 37.6 °C on admission. Physical examination revealed no palpable lymph nodes, and there were mild macular rashes on the trunk and both thighs but no eschar. The patient stated that the skin lesions tended to improve 1–2 days before admission. Mild tenderness of the right upper quadrant of the abdomen was noted. She complained of febrile sensation, vomiting, and nausea. The initial complete blood count revealed leukopenia and thrombocytopenia. She had increased aspartate aminotransferase (AST), alanine transaminase, and lactate dehydrogenase (LDH) levels. The blood test findings are shown in Table 1.

Based on the patient’s outdoor activity history and laboratory findings, an SFTSV infection or other tick-borne diseases were suspected. Since SFTSV infection and other tick-borne diseases cannot be differentiated based on clinical characteristics alone, and the relevant serological test results take time, ciprofloxacin (400 mg intravenously twice a day) and doxycycline (100 mg twice a day) were administered empirically while waiting for the test results (Figure 1). The patient’s fever and vomiting gradually improved, and the white blood cell and platelet counts approached normal levels. Since the patient’s LDH remained elevated, she underwent ^18^F-FDG PET/CT on the eighth hospitalization day to exclude other malignant lymphomas. She had increased FDG uptake by the lymph nodes, including the neck, left axilla, and left supraclavicular lymph nodes (Figure 2a,b). The ^18^F-FDG PET/CT results of maximum standardized uptake values (SUVmax) of the lymph nodes were as follows: left level III lymph node (SUVmax; 4.4), left level IV (SUVmax; 8.2), left level V (SUVmax; 5.2), left supraclavicular area (SUVmax; 10.0), and left axillary lymph node (SUVmax; 25.3). Additional evaluation was conducted to exclude lymphoma secondary to enlarged axillary lymph nodes due to a high SUVmax value on ^18^F-FDG PET/CT. A core needle biopsy of lymph nodes was also performed to exclude lymphoma. The real-time RT-PCR targeting the M and S segments of SFTSV in the blood was positive on the same day. On pathologic examination, the left axillary and supraclavicular lymph nodes exhibited extensive necrosis with atypical lymphoid hyperplasia (Figure 3). These findings were similar to those of Kikuchi disease. There were no immunohistochemical staining findings suggestive of lymphoma.

As the prevalence of tick-borne infectious diseases such as Q fever, brucellosis, anaplasmosis and Lyme disease is low in South Korea [4], these were not considered during initial evaluation. Additional tests for Q fever and brucellosis were planned in cases where the patient’s clinical symptoms did not improve or the initial tests could not demonstrate any significant results. However, the results of SFTSV real-time RT-PCR on blood samples were confirmed to be positive in this case, and the patient’s condition gradually improved without steroids or antiviral agents. She was therefore discharged without further evaluation on hospital day 14.

The axillary and supraclavicular lymph nodes were of a normal size on CT one month later (Figure 4a,b). She was clinically diagnosed with necrotizing lymphadenitis caused by SFTSV. SFTSV-PCR was performed on the remaining lymphoid tissue and the test was negative. However, the tissue sample was expectedly insufficient. In this case, the patient’s ^18^F-FDG PET/CT findings and necrotizing lymphadenitis mimicked malignant lymphoma. The present study was approved by the institutional review board of Pusan National University Yangsan Hospital (IRB No. 05-2021-077).

## 3. Discussion

SFTS is an emerging infectious zoonosis in China, Japan, South Korea, and Vietnam [1,2]. SFTS begins with nonspecific prodrome, including fever, myalgia, vomiting, and diarrhea, that persists for about a week. Following the prodrome, patients may develop hemorrhagic rash, altered mental status, and multiorgan failure [2]. Laboratory findings include thrombocytopenia, elevated liver enzymes, and disseminated intravascular coagulopathy [2]. In Korea, a total of 36 cases SFTS have been confirmed since the first patient in 2013, and according to a report by the Korea Centers for Disease Control and Prevention, the number of SFTS patients gradually increase every year, with 243 cases diagnosed in 2020 [4]. Although the average case fatality rate varies among regions and years, the mean mortality rate of SFTS cases has remained relatively high in Japan (27%), South Korea (23.3%), and China (5.3–16.2%) [5].

This was a case of necrotizing lymphadenitis caused by SFTSV, which mimicked malignant lymphoma on ^18^F-FDG PET/CT. Although the lymphoid tissue was insufficient for RT-PCR for SFTSV, the clinical course, history, positive PCR for SFTSV in blood, and improvement in lymphadenitis after one month of follow-up were suggestive of necrotizing lymphadenitis caused by SFTSV infection. Only one report has documented the use of ^18^F-FDG PET/CT in a patient with SFTS. In the case of that patient, traces of tick bites were observed, and as a result of RT-PCR, SFTSV was confirmed in lymph node tissue, clarifying the diagnosis of SFTS [6]. However, as in this case, it was necessary to exclude malignant lymphoma due to a high SUV max value of lymph nodes on ^18^F-FDG PET/CT.

The enlargement of lymph nodes can be caused by several conditions, including systemic infection or inflammation, infiltration of the neoplasm, and proliferation of lymphocytes [7,8]. Patients with enlarged peripheral lymph nodes and associated systemic symptoms should be evaluated for lymphoid malignancies [9]. ^18^F-FDG PET/CT has been applied in assessing symptoms suggestive of malignant and benign diseases, such as fever of unknown origin [10,11,12].

A previous study investigated the increased SUVmax in aggressive lymphoma (SUVmax 5.9–31.3), Kikuchi-Fujimoto disease (SUVmax 6.0–19.2), and tuberculous lymphadenitis (SUVmax 3.0–23.8) on ^18^F-FDG PET/CT [13,14]. One report recorded an SUVmax of 28.7 in lymphadenitis caused by SFTSV [6]. An increase in SUVmax to 25.3 was also observed in this case. In view of these reports, the ^18^F-FDG PET/CT findings alone are insufficient to differentiate infection and malignancy [15]. ^18^F-FDG uptake is non-specific for some cancers because the ^18^F-FDG avidity is due to increased glycolysis and glucose transporter activity. A biopsy is essential to definitively distinguish malignancy from infection [15].

Further studies on the ^18^F-FDG PET/CT and pathologic findings of necrotizing lymphadenitis caused by SFTS are needed to investigate the pathogenesis of SFTS.

## 4. Conclusions

We report the second case of necrotizing lymphadenitis caused by SFTS mimicking malignant lymphoma on ^18^F-FDG PET/CT. Clinicians should consider SFTS in patients with increased SUVmax of regional lymph nodes on ^18^F-FDG PET/CT and lymph node biopsy findings of necrotizing lymphadenitis. It is also necessary to closely examine the patient’s outdoor activities and signs of tick bites.

## Figures and Tables

**Figure 1 ijerph-18-10488-f001:**
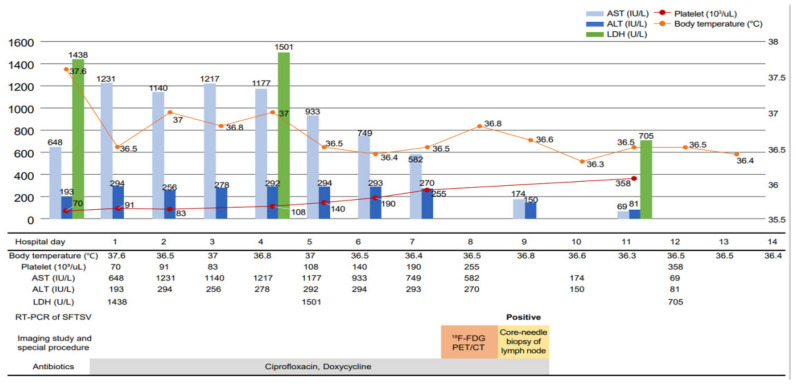
Changes in laboratory results and body temperature according to the course of hospitalization.

**Figure 2 ijerph-18-10488-f002:**
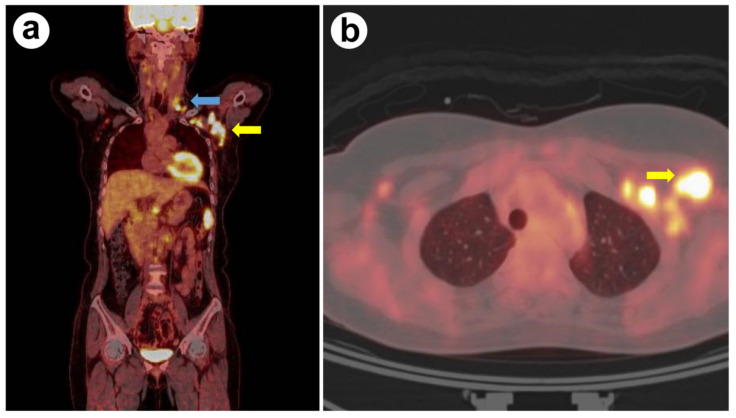
(**a**) ^18^F-FDG PET/CT findings of the patient, coronal view; blue arrow points to supraclavicular lymph node, yellow arrow points to the axillary lymph node. (**b**) ^18^F-FDG PET/CT findings of the patient, horizontal view; yellow arrow points to the axillary lymph node.

**Figure 3 ijerph-18-10488-f003:**
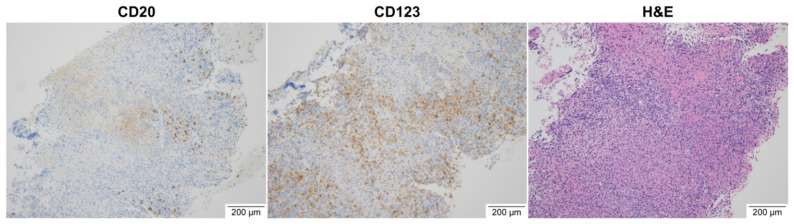
Pathologic findings of the axillary lymph node through a core needle biopsy. CD20 (a marker of B cells) is negative, and CD123 (a marker of basophils and dendritic cells) is positive. It shows extensive necrosis and atypical lymphoid hyperplasia on H&E stain.

**Figure 4 ijerph-18-10488-f004:**
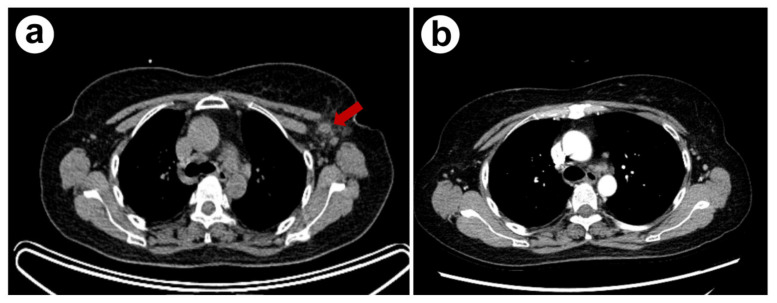
(**a**) Enlarged lymph nodes on the axillary area (red arrow) were observed detected during ^18^F-FDG PET/CT scan. (**b**) Contrast-enhanced chest CT performed one month after discharge, enlargement of axillary lymph node was improved.

**Table 1 ijerph-18-10488-t001:** Laboratory results of the case.

Tests	Results	Reference Range
WBC * (/uL) Neutrophil (/uL) Lymphocyte (/uL) Hemoglobin (g/dL) Platelet (/uL) C-reactive protein (mg/dL) ESR ^†^ (mm/h) Aspart aminotransferase, AST (IU/L) Alanine transaminase, ALT (IU/L) Lactate dehydrogenase, LDH (IU/L) Total bilirubin (mg/dL) *Orientia tsutsugamushi* antibody Hantaan virus antibody Leptospira antibody VDRL ^‡^ Anti-HIV ^§^ Anti-HCV ^∥^ Anti-HAV ** IgM Anti-HBs Ab/HBs Ag ***	1590 970 380 11.9 70,000 0.39 15 648 193 1483 0.3 Negative Negative Negative Negative Negative Negative Negative Negative/negative	4000–11,000 1700–7000 1000–4000 13.5–17.5 140,000–400,000 0–0.5 0–20 0–35 0–35 273–490 0.3–1.2 Negative Negative Negative Negative Negative Negative Negative Negative/negative

^1^ Abbreviations: WBC *, white blood cell; ESR ^†^ erythrocyte sedimentation rate; VDRL ^‡^ venereal disease research laboratory; anti-HIV ^§^, anti-human immunodeficiency virus; Anti-HCV ^∥^, anti-hepatitis C virus; Anti-HAV ** IgM, anti-hepatitis A immunoglobulin M; Anti-HBs Ab/HBs Ag ***, anti-hepatitis B surface antibody/hepatitis B surface antigen.

## Data Availability

Due to privacy and ethical concerns, neither the data nor the source of the data can be made available.

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
