# Peer review of "Severe Fever with Thrombocytopenia Syndrome with Necrotizing Lymphadenitis in a Patient who Underwent 18F-FDG PET/CT: A Case Report"

_ijerph, 2021, doi:10.3390/ijerph181910488_

Round 1
Reviewer 1 Report
Dear Authors,
This is an interesting case report of a rare clinical situation. Severe fever with thrombocytopenia syndrome with necrotizing lymphadenitis has similar manifestations to malignant lymphoma on 18F-FDG PET/CT.
However, attention should be paid to the following problems. The manuscript should be subjected to a major revision.
- The abstract needs to be rewritten since the case is not well summarized and the conclusion is not clearly stated.
- The introduction mainly describes the symptoms caused by SFTSV. Consider the topic of this article, it can be better to point out that this viral disease often needs to be distinguished from other diseases in the introduction section.
- In page 2, lines 56 to 59. The authors first describe the patient is initial diagnosed as SFTSV infection. But there is a contradiction between the subsequent treatment measures and the initial diagnosis. The author should describe the diagnosis and treatment process logically.
- The discussion part is inadequate and moderate improvements are required. Again, according to the article’s topic, differential diagnosis and misdiagnosis of SFTSV infection are necessary to be the focus of the discussion. The authors should systematic review the literatures in this section. Finally, it is suggested that 18F-FDG PET/CT is unable to distinguish malignant lymphoma and necrotizing lymphadenitis caused by SFTSV.
Best regards
Author Response
Reviewer 1:
- The abstract needs to be rewritten since the case is not well summarized and the conclusion is not clearly stated.
Response:
Thank you for this comment. The abstract has been modified as follows. (lines 12-19, page 1)
Severe fever with thrombocytopenia syndrome (SFTS), also known as fever, thrombocytopenia, and leukopenia syndrome, is an emerging zoonotic disease in Asian countries. SFTS should be suspected in patients presenting with fever, thrombocytopenia, and leukopenia and have a history of tick exposure in an endemic area. Since SFTS can be accompanied by lymphadenopathy, particularly generalized lymphadenopathy, it can be confused with a primary malignancy, such as lymphoma. The study reports a case of SFTS accompanied by multiple lymphadenopathies, which mimicked malignant lymphoma on F-18 fluorodeoxyglucose positron emission tomography/computed tomography.
- The introduction mainly describes the symptoms caused by SFTSV. Consider the topic of this article, it can be better to point out that this viral disease often needs to be distinguished from other diseases in the introduction section.
Response:
Thank you for the recommendation. We revised the Introduction as follows (lines 23-33, page 1)
Severe fever with thrombocytopenia-causing phlebovirus has been officially named Dabie bandavirus and belongs to the genus bandavirus, family Phenuiviridae, and order bunyavirales [1,2]. This virus is also known as the severe fever with thrombocytopenia syndrome virus (SFTSV). Severe fever with thrombocytopenia syndrome (SFTS) is endemic to China, South Korea, Japan, and Vietnam [1–3]. Patients with SFTS are usually infected with SFTSV through tick bites [1]. Since SFTS can be accompanied by lymphadenopathy, particularly generalized lymphadenopathy, it can be confused with a primary malignancy, such as lymphoma. We report a case of SFTS accompanied by multiple lymphadenopathies, which mimicked malignant lymphoma on F-18 fluorodeoxyglucose positron emission tomography/computed tomography (18F-FDG PET/CT).
- In page 2, lines 56 to 59. The authors first describe the patient is initial diagnosed as SFTSV infection. But there is a contradiction between the subsequent treatment measures and the initial diagnosis. The author should describe the diagnosis and treatment process logically.
Response:
Thank you for the thoughtful comments.
In Korea, SFTSV infection needs to be differentiated from scrub typhus, anaplasmosis, leptospirosis, and Hantaan virus infection. However, in this case, anaplasmosis could not be ruled out because there was no commercially available test except peripheral blood smear exam in Korea. For the treatment of anaplasmosis and Scrub typhus, ciprofloxacin and doxycycline were empirically administered to the patient.
We have added the following sentence (lines 57-62, page 2)
Because SFTSV infection and other tick-borne diseases cannot be differentiated based on clinical characteristics alone, and the relevant serological test results take time, ciprofloxacin (400 mg intravenously twice a day) and doxycycline (100 mg twice a day) were administered empirically while waiting for the test results (Figure 1). The patient’s fever and vomiting gradually improved, and the white blood cell and platelet counts approached normal levels.
- The discussion part is inadequate and moderate improvements are required. Again, according to the article’s topic, differential diagnosis and misdiagnosis of SFTSV infection are necessary to be the focus of the discussion. The authors should systematic review the literatures in this section. Finally, it is suggested that 18F-FDG PET/CT is unable to distinguish malignant lymphoma and necrotizing lymphadenitis caused by SFTSV.
Response:
Thank you for the thoughtful comments. We added additional information to the Discussion as follows (lines 103-126, pages 4 to page 5)
This was a case of necrotizing lymphadenitis caused by SFTSV, which mimicked malignant lymphoma on 18F-FDG PET/CT. Although the RT-PCR for SFTSV in lymphoid tissues was negative, the clinical course, history, positive PCR for SFTSV in blood, and improvement in lymphadenitis after one month of follow-up were suggestive of necrotizing lymphadenitis caused by SFTSV infection. Only one report has documented the use of 18F-FDG PET/CT in a patient with SFTS [8]. Similar to this case, it was necessary to exclude malignant lymphoma in this case due to a high SUV max value of lymph nodes on 18F-FDG PET/CT.
Enlargement of lymph nodes can be caused by several conditions, including systemic infection or inflammation, infiltration of the neoplasm, and proliferation of lymphocytes [9, 10]. Patients with enlarged peripheral lymph nodes and associated systemic symptoms should be evaluated for lymphoid malignancies [11]. 18F-FDG PET/CT has been applied in assessing symptoms suggestive of malignant and benign diseases, such as fever of unknown origin [12–14].
A previous study investigated the increased SUVmax in aggressive lymphoma (SUVmax 5.9-31.3), Kikuchi-Fujimoto disease (SUVmax 6.0-19.2), and tuberculous lymphadenitis (SUVmax 3.0-23.8) on 18F-FDG PET/CT [15,16]. One report recorded an SUVmax of 28.7 in lymphadenitis caused by SFTSV [8]. An increase in SUVmax to 25.3 was also observed in this case. In view of these reports, the 18F-FDG PET/CT findings alone are insufficient to differentiate infection and malignancy [7]. 18F-FDG uptake is non-specific for some cancers because the 18F-FDG avidity is due to increased glycolysis and glucose transporter activity. A biopsy is essential to definitively distinguish malignancy from infection [7].
Reviewer 2 Report
Concise, well-described manuscript.
Did the patient have a history of tick and/or ectoparasite bites? This should be included in the case report, especially considering that Orientia was considered as a causative agent.
Author Response
Reviewer 2 :
- Did the patient have a history of tick and/or ectoparasite bites? This should be included in the case report, especially considering that Orientia was considered as a causative agent.
Response:
Thank you for the thoughtful comments. The patient had no memory of insect or tick bites. When she visited our hospital, she had a slight macular rash on trunk and both thighs. We were unable to photograph these skin lesions during initial admission. However, these lesions were faint and considerably different from the maculopapular rash observed in scrub typhus. In addition, she reported improvement of the skin lesion 1–2 days before admission.
The reasons for administering ciprofloxacin and doxycycline even though the skin lesions were not typical of scrub typhus are as follows.
- Other clinical manifestations and laboratory findings were similar to those of other rickettsial diseases, including scrub typhus, Q fever, anaplasmosis, and ehrlichiosis.
- Empirical administration of ciprofloxacin and doxycycline while awaiting the test results would be beneficial to the patient. Because the results of PCR for SFTSV take at least 3 days, including the delivery time, and it is not possible to test for all suspected rickettsial diseases, empirical therapy may be necessary.
We revised the manuscript as follows (lines 37-46, page 1 to page 2)
She had climbed a mountain about two weeks prior to our hospital admission and had no memory of insect or tick bite. The patient was admitted to another hospital due to fever after 7 days of climbing. ~The patient was alert with a blood pressure of 90/60 mmHg, pulse of 84 beats/min, and body temperature of 37.6℃ on admission. Physical examination revealed no palpable lymph nodes, and there were mild macular rashes on the trunk and both thighs but no eschar. The patient stated that the skin lesions showed improvement 1–2 days before admission.

Reviewer 3 Report
The manuscript describes a patient with SFTS. The clinical presentation included fever, nausea, vomiting, leukocytopenia, thrombocytopenia, elevated AST/ALT, elevated LDH, and hepatosplenomegaly. 18F-FDG PET/ CT found necrotizing lymphadenitis, but the biopsy denied lymphoma. SFTSV was detected by RT-qPCR.
This is the second reported case of necrotizing lymphadenitis caused by SFTSV, proving us a better understanding of the pathogenesis of SFTSV infection. My major question is that the authors did not rule out other rickettsiae infections. Generally the manuscript is well-written, but small mistakes are found.
Major points:
. P2 Table 1 & L57-60: Only mite-borne scrub typhus was tested, but tick-borne rickettsioses were not considered. The patient was treated with Ciprofloxacin along with doxycycline and responded well. This could be a sign of rickettsioses. Even SFTSV was detected in blood by RT-qPCR, rickettsioses should not be ruled out without further examination.
. P1 L28: There is only one human case of SFTS reported in Taiwan, why did the authors believe the disease is endemic?
Minor points:
. Title: whether to use the capital letters should be consistent, and the “18” should be superscript.
. Did the patient recall any tick exposure?
. P4 L94: A period is missing in the end of the sentence.
. P4 L104-105: Please check grammar errors.
. P4 L108: A period is missing in the end of the sentence.
. P5 L121: The “18” should be superscript. Please check grammar errors.
Author Response
Reviewer 3 :
- P2 Table 1 & L57-60: Only mite-borne scrub typhus was tested, but tick-borne rickettsioses were not considered. The patient was treated with Ciprofloxacin along with doxycycline and responded well. This could be a sign of rickettsioses. Even SFTSV was detected in blood by RT-qPCR, rickettsioses should not be ruled out without further examination.
Response:
-Thank you for these important comments. We think that tests for Q fever, brucellosis, and anaplasmosis should have been performed. However, these tests are not widely performed in South Korea because of the low prevalence of the diseases. We had planned to take the sample to the National Institute of Health of Korea for Q fever and Brucella testing in case the SFTSV test was negative. However, the PCR test for Anaplasma phagocytophilum was not available commercially in Korea.
For the above-stated reasons, we did not conduct additional tests because it is unlikely that the patient was infected with both the virus and rickettsiae at the same time, and there were no abnormal findings in the follow up examination after one month.
- P1 L28: There is only one human case of SFTS reported in Taiwan, why did the authors believe the disease is endemic?
Response:
Thank you for the comment. We apologize for the error in the literature review.
Severe fever with thrombocytopenia syndrome (SFTS) is endemic to China, South Korea, Japan, and Vietnam [1–3]. (lines 27-28, page 2)
. Title: whether to use the capital letters should be consistent, and the “18” should be superscript.
Response:
- We have corrected the casing in the title to be consistent. Thank you.
. Did the patient recall any tick exposure?
Response
Thank you for this comment. The patient had no memory of any insect or tick bite.
Lines 37-38, page 2
She had climbed a mountain about two weeks prior to our hospital admission and had no memory of insect or tick bite
. P4 L94: A period is missing in the end of the sentence.
Response:
Thank you. We have added the missing period
Lines 95-97, page 4
The present study was approved by the institutional review board of Pusan National University Yangsan Hospital (IRB No. 05-2021-077).
. P4 L104-105: Please check grammar errors.
Response:
Thank you for the comment. We have revised the sentence accordingly.
In the process of editing the manuscript, the sentence was deleted.
(lines 103-111, page 4 to 5)
This was a case of necrotizing lymphadenitis caused by SFTSV, which mimicked malignant lymphoma on 18F-FDG PET/CT. Although the RT-PCR for SFTSV in lymphoid tissues was negative, the clinical course, history, positive PCR for SFTSV in blood, and improvement in lymphadenitis after one month of follow-up were suggestive of necrotizing lymphadenitis caused by SFTSV infection. Only one report has documented the use of 18F-FDG PET/CT in a patient with SFTS [8]. Similar to this case, it was necessary to exclude malignant lymphoma in this case due to a high SUV max value of lymph nodes on 18F-FDG PET/CT.
. P4 L108: A period is missing in the end of the sentence.
Response:
Thank you. We have added the missing period (lines 105-108, page 4)
Although the RT-PCR for SFTSV in lymphoid tissues was negative, the clinical course, history, positive PCR for SFTSV in blood, and improvement in lymphadenitis after one month of follow-up were suggestive of necrotizing lymphadenitis caused by SFTSV infection.
. P5 L121: The “18” should be superscript. Please check grammar errors.
Response:
Thank you. We have revised the sentence to correct this. (line 127, page 5), We checked the grammar again. If there are any error we’ve missed, please let us know, and we willl take them into account
